# CXCL12 and CXCR4 as Novel Biomarkers in Uric Acid-Induced Inflammation and Patients with Gouty Arthritis

**DOI:** 10.3390/biomedicines11030649

**Published:** 2023-02-21

**Authors:** Seong-Kyu Kim, Jung-Yoon Choe, Ki-Yeun Park

**Affiliations:** 1Division of Rheumatology, Department of Internal Medicine, Catholic University of Daegu School of Medicine, 33, Duryugongwon-ro 17-gil, Nam-gu, Daegu 42472, Republic of Korea; 2Arthritis and Autoimmunity Research Center, Catholic University of Daegu, 33, Duryugongwon-ro 17-gil, Nam-gu, Daegu 42472, Republic of Korea

**Keywords:** gout, CXCL12, CXCR4, uric acid

## Abstract

The aim of this study was to evaluate the expression of chemokine receptor CXCR4 and its ligand CXCL12 in patients with gout and uric acid-induced inflammation. A total of 40 patients with intercritical gout and 27 controls were consecutively enrolled. The serum levels of interleukin-1β (IL-1β), IL-18, CXCL12, and CXCR4 were assessed using an enzyme-linked immunosorbent assay. The gene and protein expressions for these target molecules were measured in human U937 cells incubated with monosodium urate (MSU) crystals using a real-time reverse transcription polymerase chain reaction and Western blot analysis. Patients with intercritical gout showed higher serum IL-1β, IL-18, and CXCL12 levels, but not the serum CXCR4 level, than in the controls.The serum CXCR4 level in gout patients was associated with the serum IL-18 level, uric acid level, and uric acid/creatinine ratio (*r* = 0.331, *p* = 0.037; *r* = 0.346, *p* = 0.028; and *r* = 0.361, *p* = 0.022, respectively). U937 cells treated with MSU crystals significantly induced the CXCL12 and CXCR4 mRNA and protein expression in addition to IL-1β and IL-18. In cells transfected with IL-1β siRNA or IL-18 siRNA, the CXCL12 and CXCR4 expression was downregulated compared with the non-transfected cells in MSU crystal-induced inflammation. In this study, we revealed that CXCL12 and CXCR4 were involved in the pathogenesis of uric acid-induced inflammation and gouty arthritis.

## 1. Introduction

Gout is a chronic inflammatory disease triggered by an excessive deposition of monosodium urate (MSU) in articular and extra-articular structures, usually in subjects with hyperuricemia [1]. It initially manifests as acute inflammatory arthritis and progresses to chronic synovitis, accompanied by tophi depositions and joint damage. Although the pathogenesis of gout has not been determined, the activation of the NLRP3 inflammasome triggered by uric acid is considered to be a key pathogenic mechanism in the acute inflammatory response of gout, which leads to the production of proinflammatory cytokines, including interleukin-1β (IL-1β) and IL-18 [2,3].

Chemokines are a family of small, secreted chemotactic cytokines that play a crucial role in the stimulation of leukocyte migration and adhesion to inflammatory lesions through interactions with G protein-coupled chemokine receptors. They are involved in cellular homeostasis, immune system activation, and inflammatory responses [4]. Chemokines and their receptors have been implicated in the pathogenesis of inflammation of diverse joint diseases and have also been considered as potential therapeutic targets [5]. In addition, the enhanced production of chemokines—including IL-8 (C-X-C motif chemokine ligand 8 (CXCL8)) [6], macrophage inflammatory protein (MIP)-1α (C-C motif chemokine ligand (CCL3)) [7], MIP-2/CXCL2 [7], monocyte chemoattractant protein-1 (MCP-1) [8], and CXCL16 [9]—has been found to be associated with the pathogenesis of uric acid-induced inflammation in gouty arthritis.

CXCL12, formerly known as stromal-derived factor-1 (SDF-1), is produced in a broad variety of human tissues (including skin, kidney, colon, and joint tissues) as well as in various cell types (including stromal cells, monocytes, and synovial cells) [10,11]. C-X-C chemokine receptor 4 (CXCR4) is a chemokine receptor in the G protein-coupled receptor (GPCR) superfamily [10,11]. CXCR4 binds to its ligand CXC chemokine CXCL12 and plays a role in the induction of the recruitment of inflammatory cells, including leukocytes and endothelial cells [12]. The CXCL12/CXCR4 axis has been implicated in the pathogenesis of several types of inflammatory arthritis and autoimmune diseases, including rheumatoid arthritis (RA) [13,14], osteoarthritis (OA) [15,16], systemic lupus erythematosus (SLE) [17], and ankylosing spondylitis (AS) [18]. Evidence of the pathogenic role of CXCR4 and CXCL12 in uric acid-induced inflammation has not been presented. Thus, the aim of this study was to compare the CXCR4 and CXCL12 levels between gout patients and controls and to determine the role of these molecules in uric acid-induced inflammation.

## 2. Materials and Methods

### 2.1. Study Population

In this study, we consecutively enrolled male patients with intercritical gout (*n* = 40) who were older than 18 years and met the classification criteria of gout proposed by the American College of Rheumatology/European League Against Rheumatism [19]. Patients with gout were those who continued to receive uric acid-lowering agents and who had no history of gout attacks in the previous month. Age- and gender-matched controls (*n* = 27) also were recruited. We excluded those with any medical history or treatment for other inflammatory arthritis or autoimmune disease-related arthritis (including RA, AS, pseudogout, psoriatic arthritis, SLE, or Sjögren’s syndrome) through a review of the medical records or an interview with each subject. The clinical data (including the age, gender, blood urea nitrogen, creatinine, estimated glomerular filtration rate (eGFR), uric acid, C-reactive protein (CRP), and erythrocyte sedimentation rate (ESR)) of the gout patients were collected at the time of study enrollment (Table 1). The estimated glomerular filtration rate (eGFR) was calculated by the original Modification of Diet in Renal Disease (MDRD) equation using the serum creatinine, age, and gender, as follows: MDRD eGFR (mL/min/1.73 m^2^) = 175 × (serum creatinine)^−1.154^ × (age)^−0.203^. The current medication of the gout patients was identified, and included febuxostat, allopurinol, benzbromarone, colchicine, steroids, non-steroidal anti-inflammatory drugs, and diuretics.

We collected venous blood samples from the gout patients and controls, placed them in tubes for the serum separation, and centrifuged them at 2500 rpm for 5 min (NeoGenesis Co., Ltd., Seoul, Korea) to obtain the supernatant. Each supernatant was collected into an Eppendorf tube and stored in a freezer at −80 °C until the ELISA analysis.

### 2.2. Cell Culture and Preparation of MSU Crystals

Cells from the human monocytic leukemia cell line U937 were obtained from the Korean Cell Line Bank (KCLB, Seoul, Korea) and maintained in an RPMI 1640 medium (Gibco Laboratories, Grand Island, NY, USA) supplemented with 10% fetal bovine serum (FBS) and PenStrep (100 U/mL penicillin and 100 μg/mL streptomycin) at 37 °C in a 5% CO_2_ humidified incubator. The U937 cells were differentiated by 100 ng/mL of phorbol 12-myristate 13-acetate (PMA, Sigma-Aldrich, St. Louis, MO, USA) and allowed to adhere for 24 h. After this, the adherent cells were treated with MSU crystals.

The MSU crystals were prepared as described in our previous study [20]. The endotoxin assay for the MSU crystals was performed using a ToxinSensor^TM^ Chromogenic LAL endotoxin assay kit (Genscript, Piscataway, NJ, USA).

### 2.3. Enzyme-Linked Immunosorbent Assay (ELISA)

The serum concentrations of IL-1β, IL-18, and CXCL12 were measured by an ELISA kit, according to the assay instructions (R&D Systems, Minneapolis, MN, USA). Briefly, 96-well plates were coated with 100 μL of each captured antibody and kept at room temperature overnight. The plates were washed three times in a wash buffer (0.05% Tween 20 in phosphate buffered saline (PBS)) and blocked with 300 μL per well of blocking reagents (1% bovine serum albumin (BSA) in PBS) for 1 h at room temperature. After washing, the serum and standards were added to each well and incubated for 2 h at room temperature. The serum IL-1β, IL-18, and CXCL12 levels were visualized with a biotin-conjugated detection antibody, followed by horseradish peroxidase-labelled streptavidin. The reaction was stopped by the addition of 2N H_2_SO_4_ to each well, and the optical density was determined on a microplate reader set to 450 nm. The serum level of CXCR4 was measured by ELISA kits (MyBioSource, San Diego, CA, USA), according to the manufacturer’s instructions.

The sera were pre-coated onto 96-well plates and incubated for 2 h at 37 °C. The plates were aspirated and a biotin antibody was added to each well and incubated for 1 h at 37 °C. After 3 washes, 100 μL of HRP-avidin (1X) was added to the wells and incubated for 1 h at 37 °C. A tetramethyl benzidine substrate was added to each well and incubated at 37 °C in the dark for 15–30 min. The reaction was stopped by the addition of 50 μL of a stop solution to each well, and the absorbance was measured at 540 nm using an ELISA plate reader (BMG Lab Technologies, Offenburg, Germany).

### 2.4. Real-Time Reverse Transcription Polymerase Chain Reaction (RT-PCR)

The cells (2 × 10^4^) were seeded in 24-well plates and treated with various concentrations of MSU crystals (0.1, 0.2, and 0.3 mg/mL) for 24 h. The total RNA was extracted using a TRIzol reagent, and complementary DNA (cDNA) was synthesized using a ReverTra Ace-α-reverse transcriptase kit (Toyobo, Osaka, Japan). The cDNA was then analyzed by real-time RT-PCR (Bio-Rad iQ5 Real-Time PCR System, Bio-Rad, Hercules, CA, USA) using an SYBR Green Mix kit (Toyobo, Osaka, Japan). The PCR amplification consisted of an initial denaturation at 95 °C for 15 min, followed by 40 cycles of 9 °C for 5 s, 55–65 °C for 30 s, and 72 °C for 15 s. The relative expression of each gene was analyzed using the ΔΔCT method.

### 2.5. Western Blot Analysis

The cells (2 × 10^6^) were seeded on 100 mm culture dishes and treated with MSU crystals (0.1, 0.2, and 0.3 mg/mL) for 24 h. The total proteins were extracted from the cells with a radioimmunoprecipitation assay buffer containing a protease inhibitor cocktail (Thermo Fisher Scientific, Rockford, IL, USA), incubated on ice for 10 min, and centrifuged at 13,000 rpm for 10 min at 4 °C. The protein concentration of the supernatant was measured with a Pierce™ BCA Protein assay kit using a microplate reader at 562 nm. Equal amounts of protein (50 μg) were separated with 10–13% SDS-PAGE gel electrophoresis and transferred to nitrocellulose membranes (Bio-Rad, Hercules, CA, USA).

After blocking with 5% BSA, the samples were probed with appropriate primary antibodies and incubated overnight at 4 °C. The membranes were subsequently reacted with the appropriate HRP-conjugated secondary antibodies (Santa Cruz Biotechnology, Santa Cruz, CA, USA) for 1 h at room temperature. The proteins were enhanced with ECL chemiluminescent detection system reagents. Images were obtained using a ChemiDoc TM XRS system (Bio-Rad).

### 2.6. Transfection of siRNA

The cells (1 × 10^4^ cells/well) were seeded in 24-well plates and transfected with 50 nM human IL-1β siRNA (HSS105299) and human IL-18 siRNA (HSS105407) using a lipofectamine RNAi MAX reagent (Invitrogen, Waltham, MA, USA) in Opti-MEM media (Gibco Laboratories). After 48 h of transfection, the cells were harvested for the real-time RT-PCR and Western analysis.

### 2.7. Statistical Analysis

The data were presented as the median and interquartile range (IQR) for the quantitative variables and numbers with percentages for the qualitative variables. The normality of the data not showing a normal distribution was verified using the Kolmogorov–Smirnov test. The comparison of IL-1β, IL-18, CXCL12, and CXCR4 between the gout patients and the controls was performed using the Mann–Whitney U test. The correlation among IL-1β, IL-18, CXCL12, CXCR4, and the laboratory variables was evaluated by Spearman’s correlation analysis. The comparison of the mRNA expression in IL-1β, IL-18, CXCL12, and CXCR4 between the cells treated with MSU crystals and the non-treated cells was assessed by a Mann–Whitney U test. A *p*-value < 0.05 was considered to be statistically significant. The statistical analysis was performed using SPSS version 19.0 (SPSS Inc., Chicago, IL, USA). The curve generation for the figures in this study was performed using GraphPad Prism 5.0 (GraphPad Software, Inc., San Diego, CA, USA).

## 3. Results

### 3.1. Comparison of Expression of CXCR4, CXCL12, IL-1β, and IL-18 between Patients with Gout and Controls

The baseline characteristics of the study population are shown in Table 1. The gout and controls were all male subjects. The median values of age between the two groups had no statistical difference (*p* = 0.349) (Table 1). IL-1β and IL-18 are crucial proinflammatory cytokines that play a role in the pathogenesis of gout [2,3]. First, we compared the levels of the two proinflammatory cytokines in the serum of the gout patients and controls (Figure 1A). The serum IL-1β and IL-18 levels in the gout patients were significantly higher than in the controls (7.31 ± 11.84 vs. 0.46 ± 0.91, *p* = 0.004 and 100.51 ± 47.63 vs. 68.49 ± 32.55, *p* = 0.002, respectively). In a comparison of CXCR4 and its ligand CXCL12, the gout patients showed a larger increase in the expression of the serum CXCL12 level compared with the controls (392.27 ± 247.54 vs. 243.90 ± 184.03, *p* = 0.010). However, there was no difference in the serum CXCR4 level between the two groups (45.45 ± 18.46 vs. 47.45 ± 8.35, *p* = 0.550).

**Table 1 biomedicines-11-00649-t001:** Baseline characteristics of the study population.

Variables	Gout (*n* = 40)	Controls (*n* = 27)
Sex, male (*n*, %)	40 (100.0)	27 (100.0)
Age (years)	63.0 (54.3, 70.0)	65.0 (51.8, 70.0)
Blood urea nitrogen (mg/dL)	19.4 (12.8, 27.1)	
Creatinine (mg/dL)	1.0 (0.8, 1.4)	
eGFR (mL/min/1.73 m^2^)	77.7 (52.4, 100.1)	
Erythrocyte sedimentation rate (mm/h)	11.5 (7.0, 21.8)	
C-reactive protein (mg/L)	0.9 (0.6, 1.9)	
Uric acid (mg/dL)	4.8 (4.2, 6.6)	
Uric acid/creatinine	5.12 (3.65, 6.59)	
Medication (*n*, %)		
Febuxostat	30 (75.0)	
Allopurinol	3 (7.5)	
Benzbromarone	5 (12.5)	
Colchicine	16 (40.0)	
Steroids	6 (15.0)	
Non-steroidal anti-inflammatory drugs	8 (20.0)	
Diuretics	2 (5.0)	

Data were described as median and interquartile range (IQR). eGFR: estimated glomerular filtration rate.

### 3.2. Correlations of IL-1β, IL-18, CXCL12, and CXCR4 with Laboratory Variables in Gout Patients

We evaluated the relationship among IL-1β, IL-18, CXCL12, CXCR4, and the laboratory variables in the patients with gout. The serum IL-1β level was not statistically associated with the serum CXCR4 and CXCL12 levels (Figure 1B). The serum IL-18 level was positively correlated with the serum CXCR4 level (*r* = 0.331, *p* = 0.037), but not the serum CXCL12 level.

There was a significant correlation between the serum CXCR4 level and uric acid (*r* = 0.346, *p* = 0.029) (Figure 1B). After an adjustment of uric acid with creatinine, the significance of the correlation between the serum CXCR4 level and uric acid persisted (*r* = 0.361, *p* = 0.022). In addition, the serum uric acid level was not associated with the serum CXCL12 level. However, we could not discover the correlations between acute phase reactants such as CRP and ESR and the serum CXCR4, CXCL12, IL-1β, and IL-18 levels (data not shown).

### 3.3. Expression of IL-1β, IL-18, CXCL12, and CXCR4 in MSU Crystal-Stimulated U937 Cells

We then evaluated whether U937 macrophages treated with MSU crystals induced the mRNA and protein expression of IL-1β, IL-18, CXCL12, and CXCR4. MSU crystals significantly induced the mRNA expression of proinflammatory cytokines IL-1β and IL-18 in human U937 cells in a dose-dependent manner (Figure 2A). In addition, the CXCL12 and CXCR4 mRNA expression markedly increased in the macrophages treated with MSU crystals compared with the non-treated cells.

Consistently, the U937 cells incubated with MSU crystals induced cleaved forms of IL-1β and IL-18 from inactive pro-IL-1β and pro-IL-18 in a dose-dependent manner (Figure 2B). In addition, the expression of CXCL12 and CXCR4 proteins was significantly increased by a stimulation with MSU crystals.

### 3.4. Expression of CXCR4 and CXCL12 by Downregulation of Either IL-1β or IL-18 in MSU Crystal-Stimulated U937 Cells

The U937 cells transfected with IL-1β siRNA suppressed the IL-1β, CXCL12, and CXCR4 mRNA expression compared with the non-transfected cells (*p* < 0.001, *p* < 0.01, and *p* < 0.05, respectively) (Figure 3A). The activation of pro-IL-1β was markedly blocked in the U937 cells transfected with IL-1β siRNA compared with the cells transfected with the negative control (NC) siRNA (Figure 3B). The macrophages transfected with the IL-1β siRNA significantly suppressed the CXCL12 and CXCR4 expression.

Consistently, the IL-18, CXCL12, and CXCR4 mRNA expression was significantly downregulated in the U937 cells transfected with the IL-18 siRNA relative to the non-transfected cells (*p* < 0.01, *p* < 0.05, and *p* < 0.05, respectively) (Figure 3C). The U937 cells transfected with the IL-18 siRNA inhibited the activation of pro-IL-18 to IL-18, respectively, and attenuated the CXCL12 and CXCR4 expression (Figure 3D).

## 4. Discussion

The NLRP3 inflammasome is an intracellular multi-protein signaling complex that is induced through stimulations with various endogenous and exogenous pathogens such as LPS, nigericin, MSU, silica, or alum [21,22]. Accumulating evidence suggests that MSU crystals induce the recruitment and assembly of NLRP3, ASC, and pro-caspase-1, triggering NLRP3 inflammasome activation and resulting in the activation of proinflammatory cytokines (including IL-1β, IL-18, tumor necrosis factor-α (TNF-α), and IL-6), which ultimately generates uric acid-induced intra-articular inflammation in the pathogenesis of gouty arthritis [2,3,23]. In addition to these inflammatory cytokines, chemokines that play a role in the ability to recruit and infiltrate leukocytes into the intra-articular space are involved as potent mediators of the inflammatory response of gouty arthritis [5]. Previous studies have found an increased expression of various chemokines (including CXCL8, CCL3, CXCL2, and MCP-1) in stimulations with MSU crystals in experimental models of gout or in the serum of gout patients [6,7,8,9]. Based on these observations, novel chemokines and their receptors might play a crucial role in the pathogenesis of gout and may also be potent therapeutic targets.

The CXCL12/CXCR4 axis is one of the most studied C-X-C chemokine/chemokine receptor subfamilies and is implicated in a broad variety of physiological and pathological conditions, including autoimmune or inflammatory diseases, tissue regeneration, and cancerous diseases [10,11,12]. The activation of GPCRs by binding ligands induces the expression of multiple genes involved in proinflammatory cytokine activation, chemotaxis, adhesion molecules, and tissue repair [24]. There are multiple complex downstream signal transduction pathways activated by the binding of CXCL12 to CXCR4. Upon binding to the ligand, CXCR4 activation induces a dissociation of heterotrimeric G protein subunits bound to the intracellular loop of CXCR4 to lead to the activation of multiple signaling pathways. Heterotrimeric G protein subunits dissociate into G_i_ and G_βγ_ subunits. The G_βγ_ subunit induces intracellular calcium mobilization and triggers the production of phosphatidyl-inositol-3-kinase (PI3K), which results in the activation of the ARK and ERK1/2 pathways and finally stimulating NF-κB, an inducible transcription factor of inflammatory genes [25]. In addition, the G_βγ_ subunit is involved in the conversion of phosphatidylinositol 4,5-bisphosphate (PIP2) into IP3 and diacylglycerol (DAG) [26]. IP3 regulates the calcium release from the endoplasmic reticulum, modulating numerous downstream signaling transduction targets. Previous studies have implicated the CXCL12/CXCR4 axis in the pathogenesis of diverse autoimmune, inflammatory, or non-inflammatory rheumatic diseases, including RA [13,14], OA [15,16], SLE [17], and AS [18]. Considering the interactions between gout and the downstream pathways in the CXCL12/CXCR4 axis, many studies have suggested that the uric acid-induced inflammatory response with activation by NF-κB might be related to the GPCR downstream signaling pathway [27,28]. Our study first identified an increased expression of CXCR4 and CXCL12 mRNA and protein under a stimulation with MSU crystals in human U937 macrophages and also a higher level of serum CXCL12 in the gout patients compared with the controls, although the serum CXCR4 level was similar between the two groups. In contrast, Murakami et al. demonstrated that the knockdown of the G_β_ subunit, a downstream molecule of GPCRs, significantly enhanced caspase-1 activation and IL-1β release by an ATP treatment, indicating that the G_β_ subunit negatively regulated NLRP3 inflammasome activation by the inhibition of ASC oligomerization [29]. Nevertheless, the CXCL12/CXCR4 axis plays a critical role in immune-mediated or inflammatory diseases (including gouty arthritis) as a therapeutic target by modulating the inflammatory and immune response and regulating the recruitment of leukocytes.

Considering the clinical significance of CXCR4 and CXCL12 in inflammatory or autoimmune rheumatic diseases, previous studies have analyzed the relationship between disease activity or severity in patients with each disease and those biomarkers. Active RA patients showed a higher CXCR4 and CXCL12 expression in the serum and joint fluid compared with patients in remission and control groups [14]. In addition, these biomarkers were significantly associated with the ESR, CRP, and DAS28 score. Qin et al. demonstrated that the activation of the SDF-1/CXCR4 axis induced subchondral bone deterioration and articular cartilage degeneration that aggravated joint destruction in anterior cruciate ligament transection OA mice models [16]. These results suggest that CXCR4 and CXCL12 might contribute to the activity and severity of OA. In the assessment of the role of chemokines in the pathogenic process of SLE, the expression level of CXCR4 in circulating B cells from peripheral blood mononuclear cells (PBMCs) in active SLE patients was significantly higher than those in inactive patients and healthy controls [17]. Consistently, the chemotactic response of CD19^+^ B cells to CXCL12 was also markedly increased in active SLE patients compared with those with inactive SLE and the controls (*p* = 0.004 and *p* = 0.001, respectively). Aeberli et al. evaluated the regulatory effect of infliximab in patients with AS on the chemokines MCP-1 and CXCL12 and their receptors CCR2 and CXCR4 in purified CD11b^+^CD14^+^ monocytes [18]. An infliximab treatment significantly decreased the serum SDF-1 level at day 84 and day 168 in both AS and RA patients. However, the downregulation of the percentage of CXCR4^+^ monocytes was not detected during the infliximab treatment for 6 months. In the present study, we could not identify the associations between the serum CXCR4 and CXCL12 levels and acute phase reactants, including ESR and CRP. However, IL-18, one of the main proinflammatory cytokines related to the pathogenesis of gout, was significantly associated with the CXCR4 level. The weak correlation with the inflammatory markers was presumed to be partially related to selection of intercritical gout patients with a low disease activity. Due to the limited clinical data of the effects of these chemokines and their receptors on the long-term prognosis and severity of inflammatory rheumatic diseases, it is necessary to evaluate the clinical significance of these biomarkers in prospective studies.

The relationship between inflammatory cytokines and chemokines is considered to be complex. In the present study, we confirmed that the IL-1β and IL-18 expression was increased by a stimulation with MSU crystals as well as in the blood of gout patients, together with CXCR4 and CXCL12. Human U937 cells transfected with IL-1β siRNA or IL-18 siRNA showed a downregulated CXCR4 and CXCL12 mRNA and protein expression, indicating that the tight control of proinflammatory cytokines may have had an additive therapeutic effect by inhibiting the action of the activated chemokines in the inflammatory response. An IL-1β pretreatment improved the homing efficacy of mesenchymal stem cells on liver failure through an increased CXCR4 expression [30]. Binding CXCL12 to CXCR4 induces the activation of mitogen-activated protein kinase (MAPK) and NF-κB, resulting in the stimulation of a diverse gene transcription [10]. Conversely, the CXCR4 knockdown of RAW 264.7 cells showed a decreased expression in proinflammatory cytokines such as IL-6 and TNF-α by the attenuation of the MAPK and NF-κB signaling pathways [31]. Based on these observations, CXCL12 and CXCR4 might be regulated by proinflammatory cytokines such as IL-1β and IL-18.

Uric acid is the main pathogenic trigger in the pathogenesis of gout through NLRP3 inflammasome activation [2,3]. In addition, there is a well-established close relationship between uric acid and cardiovascular diseases (CVDs) [32]. The pathogenic mechanism of uric acid on CVDs has not clearly defined. Oxidative stress, insulin resistance, and impaired endothelial dysfunction induced by uric acid play a role in the development of atherosclerosis. Thus, uric acid is considered to be a crucial predictor for CVD-mediated outcomes such as CV mortality and morbidity. The activation of the CXCR4/CXCL12 axis was found to play an important role in angiogenesis as well as the homing and mobilization of progenitor cells; this then results in the development of CVDs, including myocardial ischemia and infarction [33]. Considering the relationship between uric acid and the CXCR4/CXCL12 axis presented in our study, the CXCR4/CXCL12 axis may be an important mediator in the development of uric acid-induced CVD. Ultimately, the acquisition of sufficient knowledge on CXCR4/CXCR12 is expected to be useful in the prevention and treatment of CVD.

There are a few limitations to the interpretation of our results in this study. First, the size of the study population, including the gout patients and controls, was relatively small to clarify the differences between CXCL12 and CXCR4. As the expression of CXCL12 and CXCR4 was cross-sectionally analyzed, the study was limited in its ability to longitudinally confirm the change pattern of these molecules associated with medication or other inflammatory factors. Second, the expression of CXCL12 and CXCR4 was measured in the serum extracted from whole blood. It would be more convincing to confirm the expression of these target molecules from PBMCs. However, this weakness was overcome by verifying the expression of inflammatory cytokines and chemokines through experiments using human U937 macrophages treated with MSU crystals. Third, most of the gout patients had intercritical gout without an acute gout attack. The intercritical phase in gout may be clinically important for a reduction in the risk of an acute gouty flare-up [1]. As shown in Table 1, a few patients in the intercritical phase without an acute gout attack still used anti-inflammatory drugs, including colchicine, steroids, or NSAIDs. This could suggest that the patients had an inflammatory response, even at the intercritical phase. It is, therefore, necessary to identify the extent of the inflammatory response in the intercritical phase. Finally, this study was cross-sectionally designed to define the role of CXCL12 and CXCR4 in the intercritical phase. Therefore, changes in the CXCL12 and CXCR4 expression during an acute attack could not be confirmed. Therefore, a prospective longitudinal study with a larger study population and more reasonable selection criteria is needed to produce more robust results on the significance of CXCL12 and CXCR4.

## 5. Conclusions

We found that gout patients showed a higher expression of CXCL12 and proinflammatory cytokines, including IL-1β and IL-18, than members of the control group. Additionally, this study found that the expression of CXCL12 and CXCR4 could be regulated during NLRP3 inflammasome activation stimulated by MSU crystals in U937 macrophages (Figure 4). Therefore, chemokine CXCL12 and its receptor CXCR4 might be considered to be potent therapeutic targets in uric acid-induced NLRP3 inflammasome activation in gout patients.

## Figures and Tables

**Figure 1 biomedicines-11-00649-f001:**
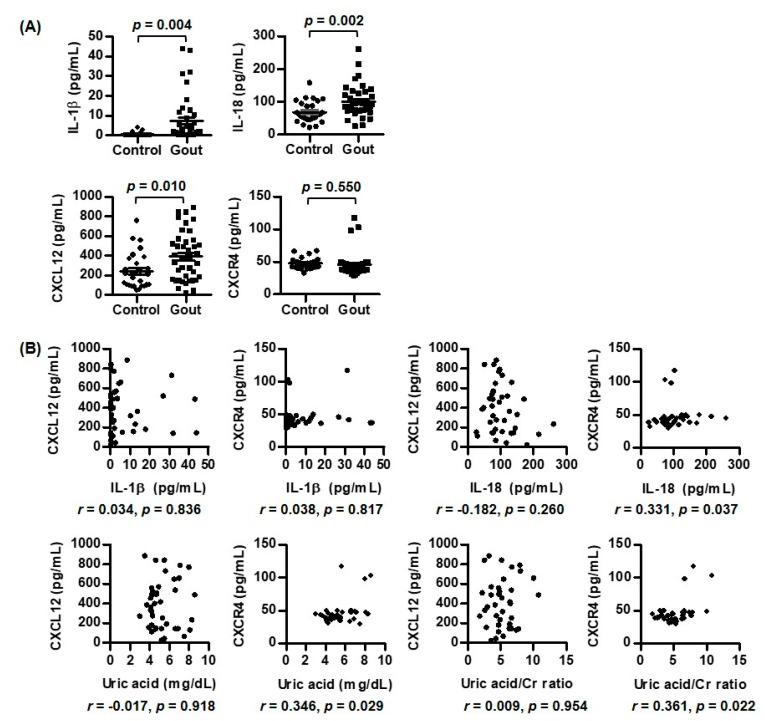
Serum IL-1β, IL-18, CXCR4, and CXCL12 expression in both gouty arthritis and controls. (**A**) Comparison of serum IL-1β, IL-18, CXCR4, and CXCL12 levels between gouty arthritis (*n* = 40) and controls (*n* = 27). (**B**) Correlation among serum IL-1β, IL-18, CXCR4, CXCL12, uric acid, and uric acid/creatinine levels in patients with gouty arthritis (*n* = 40).

**Figure 2 biomedicines-11-00649-f002:**
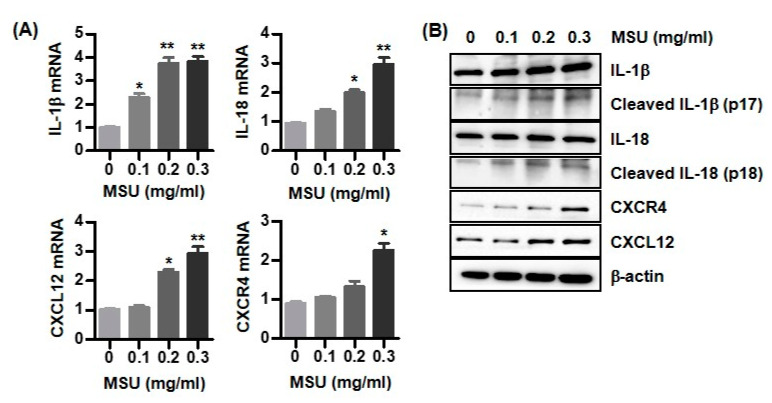
Effect of MSU crystals on expression of IL-1β, IL-18, CXCR4, and CXCL12 in U937 cells. (**A**) IL-1β, IL-18, CXCR4, and CXCL12 mRNA expression in U937 cells incubated with MSU crystals (0.1, 0.2, and 0.3 mg/mL). (**B**) IL-1β, IL-18, CXCR4, and CXCL12 protein expression in cells treated with MSU crystals. * *p* < 0.05 and ** *p* < 0.01 compared with cells without MSU crystal stimulation. The images are representative of three independent experiments.

**Figure 3 biomedicines-11-00649-f003:**
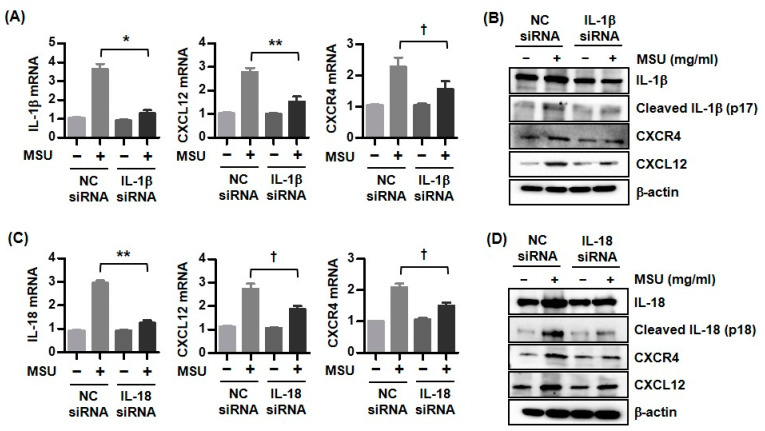
Expression of IL-1β, IL-18, CXCR4, and CXCL12 in either IL-1β siRNA- or IL-18 siRNA-transfected U937 cells treated with MSU crystals. (**A**,**B**) IL-1β, IL-18, CXCR4, and CXCL12 mRNA and protein expression in IL-1β siRNA-transfected U937 cells under stimulation of MSU crystals (0.3 mg/mL). (**C**,**D**) IL-1β, IL-18, CXCR4, and CXCL12 mRNA and protein expression in IL-18 siRNA-transfected U937 cells under stimulation of MSU crystals (0.3 mg/mL). * *p* < 0.001, ** *p* < 0.01, and ^†^
*p* < 0.05 compared with the unstimulated cells. The images are representative of three independent experiments.

**Figure 4 biomedicines-11-00649-f004:**
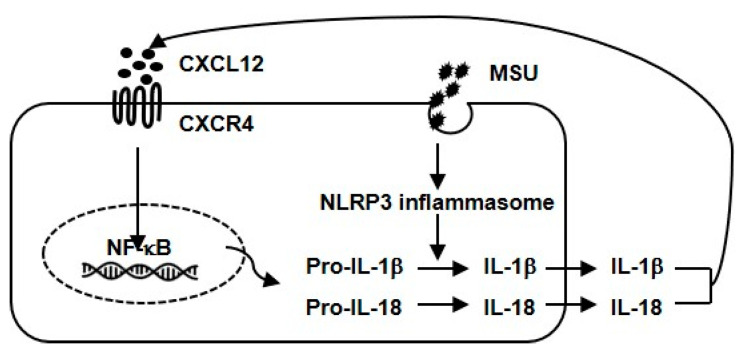
Proposed role of CXCL12 and CXCR4 in uric acid-induced inflammation. Proinflammatory cytokines such as IL-1β and IL-18 activated by MSU crystals promote CXCL12 and CXCR4 expression in macrophages, reciprocally leading to an augmentation of the inflammatory response by uric acid-induced inflammation.

## Data Availability

The data underlying this article will be shared upon reasonable request to the corresponding author.

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
