# Peer review of "CXCL12 and CXCR4 as Novel Biomarkers in Uric Acid-Induced Inflammation and Patients with Gouty Arthritis"

_biomedicines, 2023, doi:10.3390/biomedicines11030649_

Round 1

Reviewer 1 Report

Proposed paper is interesting and well written. However some revisions are needed before it can be accepted for publication:

- In the introduction or in the discussion section a few paragraph on the role of uric acid in CV disease development should be added given the importance it have in the latest years. The following review could be useful for this aspect: 10.3390/jcm10204750.

- Add in each image the number of patients per group and the number of cell used for the experiment (this was never stated in the paper).

- The ROC curve analysis should be deleted since it report a low AUC and probably (since they were not stated) low Sn and Sp values. However, also without this analysis the results are of interest. This analysis is used by the authors to reinforce the results but it seems to be that it weaken the results.

- What about patients characteristics? a table showing gout and control group characteristics should be prepared. At least age, sex, BP values, creatinine and GFR, diuretic use and other therapies should be indicated. Please also add all the variables that authors think should modify the inflammatory markers used. In fact, difference in the two population can maybe determined the founded difference in biomarkers.

- Pay particular attention to diuretic, in fact, as also recently founded, if patients were treated with diuretic higher values of uric acid will be founded and this could modify the results (10.1097/HJH.0000000000002600.). 

Author Response

Dear Editor

Manuscript ID: biomedicines-2202399

Title: CXCL12 and CXCR4 as novel biomarkers in uric acid-induced inflammation and patients with gouty arthritis

   Thank for the editor and reviewers of the ‘Biomedicines’ for reviewing our manuscript. We have made some corrections and clarifications in the revised manuscript according to the editor’s or reviewer's comments. You can find out tracing marks for changes in revised manuscript. The changes are summarized below:

Proposed paper is interesting and well written. However some revisions are needed before it can be accepted for publication:

  1. In the introduction or in the discussion section a few paragraph on the role of uric acid in CV disease development should be added given the importance it have in the latest years. The following review could be useful for this aspect: 10.3390/jcm10204750.

à Thanks for your comment. We add a new paragraph about uric acid, cardiovascular diseases, and CXCR4/CXCL12 at the end of discussion, as follows, “Uric acid is a main pathogenic trigger in the pathogenesis of gout through NLRP3 inflammasome activation [2, 3]. In addition, it has well established close relationship between uric acid and cardiovascular diseases (CVD) [32]. The pathogenic mechanism of uric acid on CVD has not clearly defined. Oxidative stress, insulin resistance, and impaired endothelial dysfunction induced by uric acid play a role in development of atherosclerosis. Thus, uric acid has been considered as a crucial predictor for CVD-mediated outcomes such as CV mortality and morbidity. Activation of CXCR4/CXCL12 axis was found to play an important role angiogenesis and homing and mobilization of progenitor cells, then results in the development of CVD including myocardial ischemia and infarction [33]. Considering the relationship between uric acid and CXCR4/CXCL12 axis presented in our study, CXCR4/CXCL12 axis may be an important mediator in the development of uric acid-induced CVD. Ultimately, acquisition of sufficient knowledge on CXCR4/CXCR12 is expected to be useful in the prevention and treatment of CVD.”.

  1. Add in each image the number of patients per group and the number of cell used for the experiment (this was never stated in the paper).

à Thanks for your kind comment. Figure 1 was generated by analysis using human blood samples, so the number of patients and control subjects was entered in the figure legend at Figure 1. Figures 2 and 3 were generated using RT-PCR and western blot, and the number of cells used for these experiments has already been described in the method section. " Cells (2 x 106) were seeded on 100-mm culture dishes and treated with MSU crystals (0.1, 0.2, and 0.3 mg/ml) for 24 h" is added for those missing in the western blot section.

  1. The ROC curve analysis should be deleted since it report a low AUC and probably (since they were not stated) low Sn and Sp values. However, also without this analysis the results are of interest. This analysis is used by the authors to reinforce the results but it seems to be that it weaken the results.

à Thanks for kind comment. We delete figure 4.

  1. What about patients characteristics? a table showing gout and control group characteristics should be prepared. At least age, sex, BP values, creatinine and GFR, diuretic use and other therapies should be indicated. Please also add all the variables that authors think should modify the inflammatory markers used. In fact, difference in the two population can maybe determined the founded difference in biomarkers.

à Thanks for your valuable comment. We generate a table 1 including general characteristics, laboratory parameters, and use of current medications of study population at the section of results. The sentence is revised as follows; “The clinical data including age, gender, blood urea nitrogen, creatinine, estimated glomerular filtration rate (eGFR), uric acid, C-reactive protein (CRP), and erythrocyte sedimentation rate (ESR) in gout patients were collected at the time of study enrollment (Table 1). Estimated glomerular filtration rate (eGFR) was calculate by the original Modification of Diet in Renal Disease (MDRD) equation using serum creatinine, age, and gender. The equation is like this: MDRD eGFR (mL/min/1.73 m2) = 175 × (serum creatinine)-1.154 × (age)-0.203. Current medication in gout patients were identified such as febuxostat, allopurinol, benzbromarone, colchicine, steroid, nonsteroidal anti-inflammatory drugs, and diuretics.”.

In addition, we add a paragraph about baseline characteristics of study population at the section of result, as follows “Baseline characteristics of study population was shown at table 1. Gout and controls were all male subjects. The median values of age between two groups has no statistical difference (p = 0.349).”.

Table 1. Baseline characteristics of study population

Variables

Gout (n = 40)

Controls (n = 27)

Sex, male (n, %)

40 (100.0)

27 (100.0)

Age (years)

63.0 (54.3, 70.0)

65.0 (51.8, 70.0)

Blood urea nitrogen (mg/dL)

19.4 (12.8, 27.1)

Creatinine (mg/dL)

1.0 (0.8, 1.4)

eGFR (mL/min/1.73m2)

77.7 (52.4, 100.1)

Erythrocyte sedimentation rate (mm/hr)

11.5 (7.0, 21.8)

C-reactive protein (mg/L)

0.9 (0.6, 1.9)

Uric acid (mg/dL)

4.8 (4.2, 6.6)

Uric acid/Creatinine

5.12 (3.65, 6.59)

Medications (n, %)

  Febuxostat 

30 (75.0)

  Allopurinol

3 (7.5)

  Benzbromarone

5 (12.5)

  Colchicine

16 (40.0)

  Steroid

6 (15.0)

  Nonsteroidal anti-inflammatory drugs

8 (20.0)

  Diuretics

2 (5.0)

Data were described as median and interquartile range (IQR).

Abbreviation: eGFR, estimated glomerular filtration rate.

  1. Pay particular attention to diuretic, in fact, as also recently founded, if patients were treated with diuretic higher values of uric acid will be founded and this could modify the results (10.1097/HJH.0000000000002600.).

à We agree about the clinical importance of diuretics and their effects on uric acid levels. However, only 2 patients were treated with diuretics. Therefore, it suggests that the possibility that diuretics can significantly affect uric acid concentration seems to be low in this study.

Reviewer 2 Report

Kim SK et al showed that gout patients showed higher expression of CXCL12 and proinflammatory cytokines including IL-1β and IL-18 than members of the control group. The study was interesting to evaluate the expression of chemokine receptor CXCR4 and its ligand CXCL12 in patients with gout and uric acid-induced inflammation. However, this study has several points for improvements.

Major comments:

1. First, it is not clear what was collected in gout patients and controls. Also, there is no description of how the blood was handled.

2. Second, it is unclear why MSU crystals were added to the cells. Since MSU crystals are not present in blood, it is unclear what is being modeled. It is unlikely that there is a need to compare IL-1β, IL-18, CXCL12, and CXCR4 levels in the blood with the environment in the joint cavity. Localization is important for uric acid research. There is a big difference whether this inflammation occurs in the blood vessels or in the joints.

3. The authors state in the manuscript that the indication criteria were "Patients with gout were those who continued to receive uric acid-lowering agents and had no history of gout attacks in the past month (page2, line 23)”. It is likely that patients with gout attacks would show more differences in inflammation values. There is some mention of the limitations of the study, but we believe it is inadequate. The authors should clearly state the reason for recruiting patients under treatment.

4. The authors were described that "the clinical data including age, gender, serum uric acid, CRP, ESR, and creatinine were collected at the time of study enrollment”. I could not find out that data in this manuscript. They should show the table for characteristic of patients.

5. I could not find any mention of where the MSU crystals were purchased from or if the patient's specimen was collected.

6. The authors demonstrated that "there was a significant correlation between serum CXCR4 level and uric acid” in Figure 1B. However, CXCR4 shows around 50 pg/ml even if UA elevates, making it difficult for the reader to recognize the correlation.

7. The authors discussed about the NLRP3 inflammasome at the beginning of the discussion chapter. Was the NLRP3 inflammasome not examined in this study, and if not, why not? In this study, it was not verified whether the activation of NLRP3 inflammasomes by stimulation of MSU crystals occurs.

Minor comments:

1. This study consecutively enrolled male patients with intercritical gout (n = 40) who 20 were older than 18 years and met the preliminary criteria for classification of gout proposed by the American College of Rheumatology/European League Against Rheumatism. What preliminary?

2. What is NC? (Page 5, line 32)

Author Response

Dear Editor

Manuscript ID: biomedicines-2202399

Title: CXCL12 and CXCR4 as novel biomarkers in uric acid-induced inflammation and patients with gouty arthritis

   Thank for the editor and reviewers of the ‘Biomedicines’ for reviewing our manuscript. We have made some corrections and clarifications in the revised manuscript according to the editor’s or reviewer's comments. You can find out tracing marks for changes in revised manuscript. The changes are summarized below:

Kim SK et al showed that gout patients showed higher expression of CXCL12 and proinflammatory cytokines including IL-1β and IL-18 than members of the control group. The study was interesting to evaluate the expression of chemokine receptor CXCR4 and its ligand CXCL12 in patients with gout and uric acid-induced inflammation. However, this study has several points for improvements.

Major comments:

  1. First, it is not clear what was collected in gout patients and controls. Also, there is no description of how the blood was handled.

à Thanks for your kind comment. We add the method for blood sampling and preparation of ELISA at the section of study population at Materials and Methods, as follows; “We collected venous blood samples from gout patients and controls, put it in each tube for serum separation, and centrifuged it at 2,500 rpm for 5 min (NeoGenesis Co., Ltd., Seoul, Korea) to obtain supernatant. Each supernatant was collected into an Eppendorf tube and stored in a deep freezer at -80 °C until ELISA analysis.”.

  1. Second, it is unclear why MSU crystals were added to the cells. Since MSU crystals are not present in blood, it is unclear what is being modeled. It is unlikely that there is a need to compare IL-1β, IL-18, CXCL12, and CXCR4 levels in the blood with the environment in the joint cavity. Localization is important for uric acid research. There is a big difference whether this inflammation occurs in the blood vessels or in the joints.

à Thanks for your valuable comment. We also partially agree with your opinion. As you well know, the most important mechanism explaining the pathogenesis of uric acid-induce inflammation or gout is the activation of the NLRP3 inflammasome, an intracellular protein complex for activating IL-1b. Of course, excessive accumulation of uric acid in the joint cavity causes an inflammatory arthritis and then finally leading to joint damages. But, macrophages and neutrophils, which are main inflammatory cells residing in the joint cavity, are primary effector cells that come into contact with uric acid at inflammation site. Many studies related to uric acid-induced inflammation have used inflammatory cells such as macrophages to understand the pathogenesis of gouty arthritis. In addition, it is considered most common to use MSU crystals to create similar stimuli such as gout in an experimental environment.

   In addition, diverse pathogenic changes including bone erosion, cartilage destruction, and synovial hyperplasia (or synovitis) are occurred in gouty arthritis. So, to study specific pathological mechanisms within the joint, specific joint resident cells including osteoclast, synovial cells, or chondrocyte must be used. In this study, it is judged appropriate to use macrophages because we wanted to confirm expression or role of inflammatory molecules such as IL-1β, IL-18, CXCL12, and CXCR4 in gouty inflammation.

  1. The authors state in the manuscript that the indication criteria were "Patients with gout were those who continued to receive uric acid-lowering agents and had no history of gout attacks in the past month (page2, line 23)”. It is likely that patients with gout attacks would show more differences in inflammation values. There is some mention of the limitations of the study, but we believe it is inadequate. The authors should clearly state the reason for recruiting patients under treatment.

à Your comment seems to be very valuable. We also in part agree with your question. Following recovery from acute gouty attack, the patient transits an asymptomatic phase of gouty arthritis. This phase is referred to as “intercritical gout.” The use of low-dose colchicine (from 0.6 mg to 1.2 mg) as prophylaxis for the prevention of gouty arthritis was common practice among rheumatologists. This study aimed to assess whether the expression of inflammatory biomarkers including IL-1β, IL-18, CXCL12, and CXCR4 in the blood of these patients was increased to confirm the clinical significance of the intercritical phase in gout. The reason for this is as follows; “Intercritical phase in gout may be clinically importance for reducing a risk for an acute gouty flare-up [1]. As shown at table 1 in present study, some patients in the intercritical phase without an acute gout attack still used anti-inflammatory drugs including colchicine, steroid, or NSAIDs. It might suggest that the patients had some inflammatory response even at the intercritical phase. It is necessary to identify the extent of inflammatory response in the intercritical phase”. It is added to study limitation in the section of discussion.

  1. The authors were described that "the clinical data including age, gender, serum uric acid, CRP, ESR, and creatinine were collected at the time of study enrollment”. I could not find out that data in this manuscript. They should show the table for characteristic of patients.

à Thanks for your valuable comment. We generate a table 1 including general characteristics, laboratory parameters, and use of current medications of study population at the section of results. The sentence is revised as follows; “The clinical data including age, gender, blood urea nitrogen, creatinine, estimated glomerular filtration rate (eGFR), uric acid, C-reactive protein (CRP), and erythrocyte sedimentation rate (ESR) in gout patients were collected at the time of study enrollment (Table 1). Estimated glomerular filtration rate (eGFR) was calculate by the original Modification of Diet in Renal Disease (MDRD) equation using serum creatinine, age, and gender. The equation is like this: MDRD eGFR (mL/min/1.73 m2) = 175 × (serum creatinine)-1.154 × (age)-0.203. Current medication in gout patients were identified such as febuxostat, allopurinol, benzbromarone, colchicine, steroid, nonsteroidal anti-inflammatory drugs, and diuretics.”.

In addition, we add a paragraph about baseline characteristics of study population at the section of result, as follows “Baseline characteristics of study population was shown at table 1. Gout and controls were all male subjects. The median values of age between two groups has no statistical difference (p = 0.349).”. 

Table 1. Baseline characteristics of study population

Variables

Gout (n = 40)

Control (n = 27)

Sex, male (n, %)

40 (100.0)

27 (100.0)

Age (years)

63.0 (54.3, 70.0)

65.0 (51.8, 70.0)

Blood urea nitrogen (mg/dL)

19.4 (12.8, 27.1)

Creatinine (mg/dL)

1.0 (0.8, 1.4)

eGFR (mL/min/1.73m2)

77.7 (52.4, 100.1)

Erythrocyte sedimentation rate (mm/hr)

11.5 (7.0, 21.8)

C-reactive protein (mg/L)

0.9 (0.6, 1.9)

Uric acid (mg/dL)

4.8 (4.2, 6.6)

Uric acid/Creatinine

5.12 (3.65, 6.59)

Medications (n, %)

  Febuxostat 

30 (75.0)

  Allopurinol

3 (7.5)

  Benzbromarone

5 (12.5)

  Colchicine

16 (40.0)

  Steroid

6 (15.0)

  Nonsteroidal anti-inflammatory drugs

8 (20.0)

  Diuretics

2 (5.0)

Data were described as median and interquartile range (IQR).

Abbreviation: eGFR, estimated glomerular filtration rate.

  1. I could not find any mention of where the MSU crystals were purchased from or if the patient's specimen was collected.

à Thanks for your valuable comment. The MSU crystals used in our experiments was manufactured in-house. Methods related to generation of MSU crystals were newly added, as follows; “MSU crystals were prepared as described in previous our study [20]. Endotoxin assay for MSU crystals was performed using the ToxinSensorTM Chromogenic LAL endotoxin assay kit (Genscript, Piscataway, NJ, USA).”.  

  1. Choe, J.Y.; Jung, H.Y.; Park, K.Y.; Kim, S.K. Enhanced p62 expression through impaired proteasomal degradation is involved in caspase-1 activation in monosodium urate crystal-induced interleukin-1b expression. Rheumatology (Oxford). 2014, 53, 1043-1053.

  1. The authors demonstrated that "there was a significant correlation between serum CXCR4 level and uric acid” in Figure 1B. However, CXCR4 shows around 50 pg/ml even if UA elevates, making it difficult for the reader to recognize the correlation.

à Thanks for your comment. We really sympathize with your concerns. However, it is rechecked using the SPSS program, but the statistical analysis is still not wrong.

  1. The authors discussed about the NLRP3 inflammasome at the beginning of the discussion chapter. Was the NLRP3 inflammasome not examined in this study, and if not, why not? In this study, it was not verified whether the activation of NLRP3 inflammasomes by stimulation of MSU crystals occurs.

à Thanks for your valuable comment. Of course, we recognize that the key inflammatory molecules evaluated in this study are CXCR4 and CXCL12. We also fully understand your concerns. Although we did not directly analyze the NLRP3 inflammasome, the generation of active IL-1b and IL-18 assayed in this study is essential for NLRP3 inflammasome activation as shown in Figure 5. Please excuse us for briefly describing the NLRP3 inflammasome at the beginning of the discussion.

Minor comments:

  1. This study consecutively enrolled male patients with intercritical gout (n = 40) who were older than 18 years and met the preliminary criteria for classification of gout proposed by the American College of Rheumatology/European League Against Rheumatism. What preliminary?

à Thanks for your kind comment. We are so sorry about the wrong description “the preliminary criteria for classification of gout proposed by the American College of Rheumatology/European League Against Rheumatism”. Therefore, we revise the sentence like this “This study consecutively enrolled male patients with intercritical gout (n = 40) who were older than 18 years and met the classification criteria of gout proposed by the American College of Rheumatology/European League Against Rheumatism [19].”.

  1. What is NC? (Page 5, line 32)

à Thanks for your kind comment. NC siRNA means “negative control siRNA”. Thus, this sentence is revised as follows, “… cells transfected with negative control (NC) siRNA (Figure 3B).”.

Reviewer 3 Report

The paper analyzes the expression co CXCR4 and CXCL12 in patients with gout and controls. Patients with gout showed higher serum IL-1b, IL-18, and CXCL12 levels. Moreover, NLRP3 inflammasome activation stimulated by MSU crystals in U937 macrophages induced induced CXCL12 and CXCR4 expression. The paper is interesting and sound. 

I have the following comments for the authors:

-        Page 2, line 20. The authors stated that patients were “with intercritical gout”. Was any patients taking drugs for gout (eg: NSAID, colchicine, other?)

-        Page 3, lines 40-46. The authors reported to have used parametric (student t-test) e non parametric test (Mann-Whitney U test). Please state how the normality of the data was assessed. 

-        Page 5, section 3.3 and 3.4. It is not clear the number of experiments performed to obtain the results. Please state the number of experiments performed.

-        Page 5, figure 2. The authors seem to have performed a Mann-Whitney U test to compare the MSU exposed cells to the ones without MSU stimulation. Please explain why that test was preferred to Kruskal Wallis test with an appropriated pairwise post test (eg Dunn test). 

Author Response

Dear Editor

Manuscript ID: biomedicines-2202399

Title: CXCL12 and CXCR4 as novel biomarkers in uric acid-induced inflammation and patients with gouty arthritis

   Thank for the editor and reviewers of the ‘Biomedicines’ for reviewing our manuscript. We have made some corrections and clarifications in the revised manuscript according to the editor’s or reviewer's comments. You can find out tracing marks for changes in revised manuscript. The changes are summarized below:

The paper analyzes the expression co CXCR4 and CXCL12 in patients with gout and controls. Patients with gout showed higher serum IL-1b, IL-18, and CXCL12 levels. Moreover, NLRP3 inflammasome activation stimulated by MSU crystals in U937 macrophages induced induced CXCL12 and CXCR4 expression. The paper is interesting and sound.

I have the following comments for the authors:

  1. Page 2, line 20. The authors stated that patients were “with intercritical gout”. Was any patients taking drugs for gout (eg: NSAID, colchicine, other?)

à Thanks for your kind comment. We generate a table 1 including general characteristics, laboratory parameters, and use of current medications of study population at the section of results.

In addition, we add a paragraph about baseline characteristics of study population at the section of result, as follows “Baseline characteristics of study population was shown at table 1. Gout and controls were all male subjects. The median values of age between two groups has no statistical difference (p = 0.349).”. 

Table 1. Baseline characteristics of study population

Variables

Gout (n = 40)

Control (n = 27)

Sex, male (n, %)

40 (100.0)

27 (100.0)

Age (years)

63.0 (54.3, 70.0)

65.0 (51.8, 70.0)

Blood urea nitrogen (mg/dL)

19.4 (12.8, 27.1)

Creatinine (mg/dL)

1.0 (0.8, 1.4)

eGFR (mL/min/1.73m2)

77.7 (52.4, 100.1)

Erythrocyte sedimentation rate (mm/hr)

11.5 (7.0, 21.8)

C-reactive protein (mg/L)

0.9 (0.6, 1.9)

Uric acid (mg/dL)

4.8 (4.2, 6.6)

Uric acid/Creatinine

5.12 (3.65, 6.59)

Medications (n, %)

  Febuxostat 

30 (75.0)

  Allopurinol

3 (7.5)

  Benzbromarone

5 (12.5)

  Colchicine

16 (40.0)

  Steroid

6 (15.0)

  Nonsteroidal anti-inflammatory drugs

8 (20.0)

  Diuretics

2 (5.0)

Data were described as median and interquartile range (IQR).

Abbreviation: eGFR, estimated glomerular filtration rate.

  1. Page 3, lines 40-46. The authors reported to have used parametric (student t-test) e non parametric test (Mann-Whitney U test). Please state how the normality of the data was assessed.

à Thanks for your valuable comment. The statistical analysis used in our study was non-parametric statistical analysis. We are sorry about the correction from Student t-test to Mann-Whitney U test. The sentence is revised as follows, “Data are presented as median and interquartile range (IQR) for quantitative variables and numbers with percentages for qualitative variables. The normality of the data was verified using the Kolmogorov-Smirnov test, not showing a normal distribution.”.

  1. Page 5, section 3.3 and 3.4. It is not clear the number of experiments performed to obtain the results. Please state the number of experiments performed.

à Thanks for your kind comment. We add “The images are representative of three independent experiments.” at section 3.3 and 3.4 figure legends.

  1. Page 5, figure 2. The authors seem to have performed a Mann-Whitney U test to compare the MSU exposed cells to the ones without MSU stimulation. Please explain why that test was preferred to Kruskal Wallis test with an appropriated pairwise post-test (eg Dunn test).

à In most experiments, the Mann-Whitney U test is applied considering individual concentrations as independent conditions (nontreated versus specific concentration), whereas the Krusakal-Wallis H test is applied as continuous conditions similarly with ANOVA test as parametric analysis. In this Figure 2, the Mann-Whitney U test was used because there was an additional intention to find the optimal concentration in the additional experiments of Figures 3 and 4.

Round 2

Reviewer 1 Report

Authors replies to all the query raised and paper improve and can now be accepted for publication.

Reviewer 2 Report

The authors have responded sincerely to the reviewers' comments. The manuscript has been revised and is considered suitable for publication.